# Molecular and Cellular Markers in Chlorhexidine-Induced Peritoneal Fibrosis in Mice

**DOI:** 10.3390/biomedicines10112726

**Published:** 2022-10-27

**Authors:** Neža Brezovec, Nika Kojc, Andreja Erman, Matjaž Hladnik, Jošt Stergar, Matija Milanič, Matija Tomšič, Saša Čučnik, Snežna Sodin-Šemrl, Martina Perše, Katja Lakota

**Affiliations:** 1Department of Rheumatology, University Medical Centre Ljubljana, 1000 Ljubljana, Slovenia; 2Faculty of Pharmacy, University of Ljubljana, 1000 Ljubljana, Slovenia; 3Faculty of Medicine, University of Ljubljana, 1000 Ljubljana, Slovenia; 4Institute of Cell Biology, Faculty of Medicine, University of Ljubljana, 1000 Ljubljana, Slovenia; 5Faculty of Mathematics, Natural Sciences and Information Technologies, University of Primorska, 6000 Koper, Slovenia; 6Reactor Physics Department, Jožef Stefan Institute, 1000 Ljubljana, Slovenia; 7Faculty of Mathematics and Physics, University of Ljubljana, 1000 Ljubljana, Slovenia; 8Department of Complex Matter, Jožef Stefan Institute, 1000 Ljubljana, Slovenia

**Keywords:** chlorhexidine gluconate, peritoneal fibrosis, mouse model, inflammation, complement

## Abstract

Understanding the tissue changes and molecular mechanisms of preclinical models is essential for creating an optimal experimental design for credible translation into clinics. In our study, a chlorhexidine (CHX)-induced mouse model of peritoneal fibrosis was used to analyze histological and molecular/cellular alterations induced by 1 and 3 weeks of intraperitoneal CHX application. CHX treatment for 1 week already caused injury, degradation, and loss of mesothelial cells, resulting in local inflammation, with the most severe structural changes occurring in the peritoneum around the ventral parts of the abdominal wall. The local inflammatory response in the abdominal wall showed no prominent differences between 1 and 3 weeks. We observed an increase in polymorphonuclear cells in the blood but no evidence of systemic inflammation as measured by serum levels of serum amyloid A and interleukin-6. CHX-induced fibrosis in the abdominal wall was more pronounced after 3 weeks, but the gene expression of fibrotic markers did not change over time. Complement system molecules were strongly expressed in the abdominal wall of CHX-treated mice. To conclude, both histological and molecular changes were already present in week 1, allowing examination at the onset of fibrosis. This is crucial information for refining further experiments and limiting the amount of unnecessary animal suffering.

## 1. Introduction

The induction of peritoneal fibrosis is a complex process. It involves peritoneal cells sensing pro-fibrotic stimuli and secreting extracellular mediators. As a result, circulating leukocytes are recruited into the peritoneum, where they play an important role in the induction and amplification of the inflammatory process and the development of fibrosis. The inflammatory response is accompanied by vasculopathy, neoangiogenesis, and the production of an abnormal amount of extracellular matrix [1].

In the past, chlorhexidine gluconate (CHX) in ethanol was used to sterilize catheters in peritoneal dialysis patients. When it was found to cause peritonitis and peritoneal sclerosis [2], it was immediately withdrawn from the practice. The CHX-induced model was first reported in rats in 1995. It was mainly established as a model of encapsulating peritoneal sclerosis [3], which is a late complication of peritoneal dialysis [2]. In 2001, CHX was used to induce the model of encapsulating peritoneal sclerosis in mice also (C57BL/6). The goal of Ishii, Y., et al. was to establish a reproducible and cost-effective model of encapsulating peritoneal sclerosis in rodents. They reported a thickening of the parietal peritoneum with the infiltration of inflammatory cells at days 3 and 7, while at days 21 and 56 fibrosis progressed markedly and the number of infiltrating immune cells regressed [4].

Since then, many researchers have used the established CHX solution (i.e., 0.1% CHX in 15% ethanol dissolved in PBS) to induce peritoneal fibrosis and test various agents. However, the CHX model is highly heterogeneous. Some researchers use CHX without alcohol. The CHX solution is usually injected intraperitoneally (ease of administration), but some researchers use a surgical procedure to insert the cannulas to avoid infection (transfer of skin microorganisms into the peritoneal cavity by repeated intraperitoneal injections) or the misconduct of intraperitoneal injection. The protocols used in the studies also differ in the number of CHX applications, which affect the duration of the study, extent of damage, inflammatory response, and pathological alterations [5,6,7,8,9,10,11,12].

Damage to the abdominal wall is the most significant consequence of CHX application in the peritoneal cavity of mice, but there are reports that other organs may also be affected. Macroscopic changes have been described in the liver and intestine, with adhesions between the peritoneum and intestinal tract and between the gastrointestinal loops [4,12,13] and the liver [13]. The effects also extend to other peritoneal organs, such as the diaphragm, stomach, spleen, and kidney, where fibrotic adhesions have also been described [3,14]. Changes in gene expression were also detected in the parietal peritoneum of model mice at the 3-week time point. Genes encoding extracellular matrix proteins (procollagens, fibronectin 1), matricellular proteins and enzymes involved in matrix remodeling (thrombospondin, tenascin C, matrix metalloproteinases, tissue inhibitors of matrix metalloproteinases, lysyl oxidases) and inflammatory molecules (certain interleukins, chemokine ligands, interferon-related genes) were markedly overexpressed in CHX-treated mice [15] as compared to controls.

Peritoneal fibrosis models are induced by infectious agents, chemical damage, or physical damage. Among chemical agents, hyperosmotic, hyperglycemic, and/or acidic solutions are used. The application of peritoneal dialysis solution (4.25% glucose) is mainly used for the studies of changes in the peritoneum during peritoneal dialysis rather than peritoneal fibrosis itself. In comparison, the degradation product of glucose detected in peritoneal dialysate, methylglyoxal, is also used to induce fibrosis and needs to be injected at a dose of 20/40 mM five consecutive days in a week for 3 weeks. However, both of the mentioned models take longer to develop fibrosis (3–7 weeks), whereas the CHX model with standard application (0.1% CHX in 15% EtOH every other day) produces peritoneal fibrosis in 1–3 weeks. The overexpression of TGF-β, a key mediator of fibrosis, by the intraperitoneal administration of adenovirus, is also applied to study peritoneal fibrosis. However, the CHX-induced model is by far the best described and most commonly used peritoneal fibrosis model in mice to study various agents [16]. The effect of the agents is usually evaluated by the histological measurement of the thickness of the parietal peritoneum. However, it has been reported that the thickness of parietal peritoneum varies in the same animal. Therefore, researchers have used various strategies, such as randomly taken multiple biopsies from the abdominal cavity or longer duration of CHX treatment. To our knowledge, the CHX-induced model has not been previously characterized histologically.

Therefore, our first aim was to characterize the model histologically to determine the location, distribution, and the type of histological alterations in the parietal and visceral peritoneum. This knowledge is the basis for the correct sampling strategy and thus for the reliability of the results obtained on the model. To further investigate the structural changes of the peritoneum, we used the hyperspectral imaging method and scanned the entire area of the parietal peritoneum in the abdomen. Our second aim was to characterize the model on a molecular and cellular basis, focusing on inflammation and fibrosis. We also wanted to determine how these processes differ between the 1-week and 3-week time points.

We show that continuous CHX treatment affects local inflammation and fibrosis over time in the parietal and visceral peritoneum (on the surface of the liver, spleen, kidney, and intestine). However, the systemic inflammatory response in this model is limited to an increase in blood polymorphonuclear cells (PMNs) ratio and does not show an increase in the acute phase proteins. The gene expression data show that all the molecular pathways are already “activated” at week 1, thus 1 week of CHX treatment is sufficient to establish a relevant model for studying the onset of fibrosis and providing opportunities for novel treatment strategies.

## 2. Materials and Methods

### 2.1. Ethical Permission

The study was conducted according to the approval for animal experiments, number U34401-13/2018/5, issued by the Veterinary Administration (Administration for Food Safety, Veterinary Sector and Plant Protection) of the Republic of Slovenia.

### 2.2. Animals and Experimental Design

Our study was performed in male C57BL/6J mice (Jackson Laboratory, Bar Harbor, ME, USA) at 16 weeks of age, weighing 23.6–30.8 g. Mice were assigned to cages with the help of randomly generated numbers to reduce possible litter effects and were part of a larger experiment using the drug treatment of this model. Mice were housed in groups of five and allowed food and water ad libitum. A 12:12 h light/dark cycle (7 a.m.–7 p.m.) was maintained and CHX applications and weighing were conducted between 7 a.m. to 9 a.m. The general well-being of the mice was assessed each day and body weight was recorded before agent administration. For each CHX-treated group, a corresponding control group treated with the vehicle PBS (Ctrl) was included as described in Table 1.

### 2.3. Induction of Peritonitis/Peritoneal Fibrosis

The induction of peritonitis/peritoneal fibrosis was performed as described by Sakai et al. [16], all the solutions were prepared fresh before each treatment. Treatment schedules, doses, and time points are summarized in Table 1. Peritoneal fibrosis was induced by intraperitoneal injections of 0.1% chlorhexidine gluconate (FujiFilm Wako Chemicals, Neuss, Germany) solution in 15% ethanol in PBS (Lonza, Basel, Switzerland) in a total volume of 200 μL every other day next to linea alba. Mice in the control group were treated with solvent only (PBS, 200 μL every other day). In addition, each mouse received daily 100 μL PBS with 1.6% ethanol as they were a part of experiment that tested an anti-fibrotic substance. Untreated mice were also included (untreated ctrl). Mice were sacrificed with CO_2_ at two different time points, the first two groups at 1 week, and the second two groups at 3 weeks of experimentation.

### 2.4. Collection of Biological Material

After euthanasia, peritoneal lavage was performed by injecting 6.5 mL of ice-cold PBS solution containing 3% fetal bovine serum (FBS) into the peritoneal cavity, gently massaging the peritoneum to dislodge any adherent cells for 30 s and collecting the lavage. Samples were centrifuged at 380 RCF for 5 min at room temperature (RT) to obtain a cell pellet, which was then resuspended in FACS buffer (2 mM EDTA, 0.5% bovine serum albumin in PBS, pH = 7.2) to the final concentration of 10^6^ cells per 100 μL.

Blood was collected from the heart immediately after euthanasia. A total of 200 μL of the blood was collected into heparin-containing tubes (16 I.U./mL) for flow cytometry. The rest was placed in a polypropylene tube and left at room temperature for 30 min to allow the blood to clot. It was then centrifuged at 2000 RCF for 10 min to obtain serum, which was then transferred to a new, clean polypropylene tube and stored at −20 °C.

The abdominal wall was harvested and cut into 5 parts (Figure 1). The first part was immediately stored in Allprotect Tissue Reagent (Qiagen, Germantown, MD, USA) for stabilization of RNA and stored at −80 °C. The second part was placed in RPMI medium (Stemcell, Vancouver, BC, Canada) for flow cytometry analysis, the third part was stored for histology, and the fourth part was stored for immunohistochemistry and further immunofluorescence labeling. For histology, spleen, kidney, liver, and intestine were also harvested.

### 2.5. Flow Cytometry

Flow cytometry was performed the same day as collection to determine cell populations in whole blood, peritoneal lavage, abdominal wall, and liver tissue. Tissue was dissociated to obtain a single-cell suspension by Liberase TM research-grade solution (1:100, Roche, Basel, Switzerland) in RPMI medium and filtration through a 0.3 μm membrane filter (Miltenyi Biotec, Bergisch Gladbach, Germany) according to the manufacturer’s protocol. Tissue cell suspension and peritoneal lavage were centrifuged at 380 RCF for 5 min at RT to obtain a cell pellet, which was then resuspended in FACS buffer to a final concentration of 10^6^ cells/mL for further analysis. Cells in the single cell suspensions were blocked with 10% mouse serum for 10 min at 4 °C to prevent unspecific binding. Cell suspension and full blood were incubated for 15 min at 4° with the combination of fluorochrome-conjugated antibodies: anti-CD45-APC-Vio770 (1:50, clone REA737), anti-Ly-6G-FITC (1:50, clone REA526) (both Miltenyi Biotec, Bergisch Gladbach, Germany), and propidium iodide solution (1 μg/mL, Miltenyi Biotec, Bergisch Gladbach, Germany). Flow cytometry was performed using MACSQuant Analyzer 10 (Miltenyi Biotec, Bergisch Gladbach, Germany), compensation was applied and cell populations with corresponding data were analyzed with FlowJo (version 10.5.3, Becton, Dickinson and Company, Ashland, OR, USA).

### 2.6. Histology

For histological assessment, samples were fixed in 4% buffered formalin-fixed for 24 h, placed in 70% ethanol until paraffin-embedded, and cut in two consecutive levels and stained according to the standard histological protocol, including haematoxylin/eosin (HE) and trichrome stain. Lesions on the abdominal wall, spleen, kidney, liver, and intestine were evaluated with light microscope Nikon Eclipse 80 (Nikon, Tokyo, Japan) by a code-blinded pathologist.

### 2.7. Immunohistochemistry

Immunohistochemistry was performed as described previously [17]. Briefly, for the preparation of cryosections, the excised pieces of the abdominal wall were fixed in 3% paraformaldehyde in PBS for 2 h at 4 °C. After overnight incubation in 30% sucrose at 4 °C, tissue was embedded in Tissue Freezing Medium (Leica Biosystems, Richmond, IL, USA), frozen, and cryosectioned with a cryostat (CM3000, Leica, Germany) into 5 μm-thick cryosections. Non-specific labeling was blocked by 3% BSA in PBS for 1 h at 37 °C. Primary rat monoclonal antibodies against SAA3 (1:100; Abcam, Cambridge, UK) and primary rabbit polyclonal antibodies against collagen IV (1:400; ab6586, Abcam, Cambridge, UK) were applied and incubated overnight at 4 °C. This was followed by rinsing in PBS and the application of appropriate secondary antibodies donkey anti-rat (1:300; Alexa Fluor^®^ 488, Invitrogen, ThermoFisher Scientific, Carlsbad, CA, USA) and goat anti rabbit (1:400; Alexa Fluor^®^ 555, Invitrogen, ThermoFisher Scientific) for 1 h at 37 °C. Manufacturers of all used antibodies provided the proof of validation on the technical specifications. Negative controls were also utilized, in which primary antibodies were replaced with PBS. Sections were mounted in a mounting medium Vectashield with DAPI (Vector Laboratories, Maravai LifeSciences, San Diego, CA, USA). Sections were observed and images were obtained with a dual bright-field and fluorescence microscope Nikon Eclipse TE300 (Nikon, Tokyo, Japan).

### 2.8. RNA Isolation

RNA from the abdominal wall was isolated using TissueLyser LT (Qiagen, Germantown, MD, USA) and RNeasy Mini Kit (Qiagen, Germantown, MD, USA) according to the manufacturer’s protocol. The quantity and quality of the RNA obtained were checked using NanoDrop 2000 spectrophotometer (Thermo Fisher Scientific, Waltham, MA, USA).

### 2.9. RNA Sequencing

The isolated RNA from the abdominal wall of 1 week and 3 weeks CHX-treated mice was used for RNA-seq library preparation with TruSeq Stranded Total RNA LT Sample Prep Kit (Gold, Ilumina, San Diego, CA, USA). Paired-end Sequencing, 2 × 100 bp, was performed using NovaSeq 6000 S4 Reagent Kit (Ilumina, San Diego, CA, USA), following the NovaSeq 6000 System User Guide Document # 1000000019358 v02 on the NovaSeq platform. For analysis, FastQC v0.11.9 was used to perform quality control analysis, followed by Trimmomatic 0.39 to remove adapter sequences.

The expression of transcripts was quantified with Salmon v1.40 using the mapping-based mode. Full decoy salmon index was prepared using the “cDNA.all” file of Ensembl release 103 and the entire genome was used as a decoy sequence. The quantification of paired-end reads was performed with options validateMappings, seqBias, and gcBias. Salmon quantification files were imported into R using tximeta (v1.8.5) package and transcript-level quantification was summarized to gene-level prior to differential gene expression analysis with DESeq2 package (v1.30.1) to compare samples from the 3 week (*n* = 3) over the 1 week (*n* = 3) experiments (*p* values were obtained with Wald test and adjusted *p* values were calculated using the Benjamini and Hochberg procedure).

In order to show the variability of samples within and between the groups with different treatments, a principal component analysis (PCA) plot was created with a variance stabilizing transformation method.

The STRING Analysis (version 11.0) of a network of upregulated differentially expressed genes in the abdominal wall between the 3-week and 1-week experimental groups was carried out to find relevant biological processes [18].

### 2.10. Quantitative Real-Time Polymerase Chain Reaction (qPCR)

A total of 1 μg of cDNA was synthesized from isolated RNA using Reverse Transcription System (Promega, Madison, WI, USA) according to manufacturer’s instructions. qPCR was performed using 5× HOT FIREPol EvaGreen qPCR Mix Plus (Solis Biodyne, Tartu, Estonia) or 2× KAPA SYBR FAST qPCR Master Mix (Roche, Basel, Switzerland) on the LightCycler 480 System (Roche). Primers were custom made and purchased from IDT (Coralville, IA, USA) with the sequences indicated in Appendix A, tested by melt curves and standard curves. Data were presented as a relative expression, normalized to GAPDH, and calculated by the 2^−∆∆CT^ method. The endogenous control expression was tested to be stable regardless of the treatment groups.

### 2.11. Enzyme-Linked Immunosorbent Assay (ELISA)

ELISA kits were used to measure the concentrations of total SAA and SAA3 (MyBioSource, San Diego, CA, USA) in serum according to the manufacturer’s instructions. Samples were diluted for measurements (1∶40 for SAA, 1:10 for SAA3).

### 2.12. Luminex Assay

G-CSF, GM-CSF, IL-1 alpha/IL-1F1, IL-1 beta/IL-1F2, IL-6, IL-10, M-CSF, TNF-alpha and uPAR were measured in serum (dilution 1:2) by magnetic bead-based multiplex assay (R&D Systems, Minneapolis, MN, USA) using the Luminex platform (MAGPIX System, Merck, Darmstadt, Germany) according to the manufacturer’s instructions. Concentrations of analytes were calculated from standard curves.

### 2.13. Hyperspectral Imaging

A custom-developed hyperspectral imaging setup [19,20,21,22], able to operate in both transmission and reflectance geometries, was used to acquire spectral and sample thickness data. In the case of the 1-week experiment, samples were imaged on a black foamed PVC substrate (Forex, Airex AG, Sins, Switzerland) and illuminated from above, recording reflectance data. In the case of the 3-week experiment, the sample was imaged on a sheet of 3 mm thick opaque white plexiglass (Acrytech, Ljubljana, Slovenia) and was illuminated from the bottom using a halogen light source (Osram, Germany), thus obtaining transmittance data. From the spectra, hemoglobin species volume fractions were extracted using the Beer-Lambert law [23,24] (3 weeks) and custom Kubelka–Munk based model [19] (1 week), employing reference absorption spectra [25] and accounting for the sample thickness. To approximate light scattering, power law [26] was used in both cases. Oxygenation and total blood volume fractions were calculated following the spectral analysis from blood species volume fraction maps.

### 2.14. Statistical Analysis

Data are presented either as mean with standard deviation or as median with interquartile range (based on normality). Correspondingly, a *t*-test or Mann–Whitney test was performed for comparison between the two groups using GraphPad Prism version 9.0. *p* values < 0.05 were considered statistically significant.

## 3. Results

### 3.1. CHX-Treated Mice Show Abdominal Pain Behavior and No Obvious Macroscopicaly Visible Changes in the Abdomen after 1 Week of Treatment

After each CHX injection, mice showed abdominal pain behavior, which was obvious for a few hours after each administration. Repeated CHX treatment induced progressive resistance to restraining (a sign of aversive event) and body weight loss (Figure 2a).

The detailed design for the whole procedure of autopsy, as well as the sampling protocol, was prepared beforehand. To characterize the location and the intensity of tissue alterations in the parietal peritoneum, the abdomen was open along the linea alba from caudal to cranial part and from linea alba to the spine area (regio lumbalis) (Figure 1), scanned by hyperspectral imaging (see section), and then fixed in the formalin for histological examination.

At autopsy, macroscopically visible changes were observed in the liver lobe edges (tumor facies visceralis of both medialis lobes, sinister and dexter) in both 1 and 3 weeks of CHX treatment. However, the changes were more obvious at 3 weeks of treatment. In addition, in the 3-week CHX treatment, one mouse developed adhesions between the intestine and abdominal wall (Figure 2b). Despite macroscopically visible changes in the liver, no significant difference was observed at relative weight of any abdominal organ (i.e., the liver, kidney, or spleen) between the untreated controls, controls, and CHX-treated mice (calculated as a ratio to whole mice weight (Figure 2c).

### 3.2. Histological Changes in Parietal and Visceral Peritoneum Start Localy and Expand after Continious Stimulus

For histological examination, the formalin-fixed abdominal wall with parietal peritoneum was cut longitudinally and crosswise to linea alba from the cranial to the caudal part of the abdomen to determine the location and pattern of the CHX-induced lesion (Figure 3).

Histological examination revealed that 1-week CHX treatment (four injections) causes injury, degradation and the loss of mesothelial cells which lead to local inflammation (edema, neutrophil, and MNC infiltration), the damage of underlying muscle layer, and extracellular matrix deposition, which progressively increases with repeating CHX treatments (eleven injections). In addition, histological examination revealed clear traces of needle injections, muscle ischemia due to vessel damage by the injection as well as bite wounds, all of which was confirmed also by HIS.

The most intense structural changes were observed in the peritoneum around the ventral parts of the abdominal wall (linea alba, regio umbilicalis) in both time points and all experimental settings. The location and pattern of histological changes was similar in both time points, only the intensity and the size of the injured area was greater after the 3-week CHX treatment. The amount of extracellular matrix deposition, including collagen fibers, was markedly higher in 3-week CHX-treated mice. Similar results were found in all experiments, which show that the CHX-induced model is robust and reproducible both in 1- and 3-week CHX treatments.

Structural changes were observed in the visceral peritoneum as well. Abdominal organs located dorsally, such as the kidney and spleen, were not affected by the 1-week CHX treatment (dislocated from the CHX solution). However, in the 3-week treatment, inflammatory infiltrate was observed in the fat tissue near the capsule of the kidney and the spleen, indicating the progressive spreading of the inflammatory process in the peritoneal cavity due to constant irritation and injury caused by CHX treatment and/or injections (infection with skin microorganisms). The 3-week CHX treatment led to the multiplying of the injury process and the progressive irruption of inflammatory cells into the parenchyma of the liver (Figure 4) and muscular layer of the intestine (Figure 5).

The histological semiquantitative evaluation of the parietal peritoneum of the abdominal wall and the visceral peritoneum that covers the liver, spleen, kidney, and intestine in the 1- and 3-weeks CHX-treated mice is shown in Table 2.

### 3.3. Use of Novel Sensitive Techniques to Evaluate Tissue Damage

Based on the hyperspectral data, we were able to calculate oxygenation data that was able to discriminate model from control subjects (Figure 6a) [24]. Similar changes were observed in oxygenation mode with CHX treatment without special differences between both 1- and 3-weeks treatments. Additionally, we have quantified the changes by determining the mode of scattering power b for subjects from the 1-week experiment (Figure 6b). This evaluation showed marked differences between healthy and diseased populations at 1-week; however, no comparison with 3-weeks was made. We have related scattering changes to changes in the tissue structure and its constituent size distribution, namely edema, fibrotic deposits, and infiltrating immune cells. Results reflect changes in the size and shape of scatters. This reflects structural changes due to the fibrotic cross-linking material, edema, or infiltration of the tissue with immune cells [19].

### 3.4. Inflammation in CHX-Treated Mice Is Present Locally, with Increased Polymorphonuclear Cells in the Blood

To quantitatively assess inflammation in CHX-treated mice, we counted PMNs systemically and locally in the abdominal wall, peritoneal lavage and liver by flow cytometry. Additionally, we measured selected inflammatory markers as mRNA expression in the abdominal wall tissue by PCR and systemic protein levels in the serum by ELISA and multiplex (MagPix).

Flow cytometry data revealed an increased median count of PMNs (represented as % of CD45+ cells) in the CHX-treated mice both locally in the abdominal wall, liver, and peritoneal lavage and systemically in the bloodstream (Figure 7a). However, only in the peritoneal lavage with long-term CHX treatment the percentage of neutrophils/PMNs significantly decreased as compared to 1 week solely, very likely due to advanced fibrosis and collagen deposition that disabled the influx of neutrophils in the peritoneal cavity.

The gene expression of inflammatory markers SAA1/SAA2, SAA3, and IL-6 (Figure 7b) was significantly increased in the abdominal wall of the CHX-treated mice compared to controls, regardless of the length of CHX treatment. The immunofluorescent imaging of the abdominal wall also confirmed cells highly positive for SAA3 protein in the CHX-treated mice (Figure 7c). However, the serum levels of the inflammatory protein markers SAA, SAA3 (Figure 7d), G-CSF, M-CSF, GM-CSF, TNFα, IL-1β, IL-6, IL-10, IL-1α, and uPAR (data not shown), were not increased in the CHX-treated mice, the last seven analytes were even below the detection limit.

### 3.5. CHX-Induced Peritoneal Fibrosis Is Characterized by the Increased Expression of Extracellular Matrix Genes and Genes That Regulate Extracellular Matrix

There was increased expression of the fibrotic markers alpha-1 type I collagen (COL1A1), alpha-smooth muscle actin (ACTA2), fibronectin (FN) (Figure 8a) and extracellular matrix (ECM) regulators, such as transforming growth factor beta 1 (TGFB1), connective tissue growth factor (CTGF), matrix metalloproteinase 3 (MMP3) and matrix metalloproteinase 14 (MMP14) (Figure 8b) in the abdominal wall of CHX-treated mice at both 1 week and 3 weeks.

### 3.6. RNA Sequencing of the Abdominal Wall Reveals High Heterogenity between Intragroup Experimental Subjects and Points out Novel Gene Expression Pathways That Change with Continous CHX Treatment

Principal component analysis (PCA) showed high intragroup heterogeneity in the clustering of individual CHX-treated samples into groups of 1-week and 3-week experiments (Figure 9a).

The RNA sequencing of the abdominal wall of CHX-treated mice revealed 49 differentially expressed genes (DEG) between the 3-week and 1-week experiments with *p* adj < 0.05 and log2FC > 0.5 or <−0.5. Among them were 38 upregulated (Figure 9b) and 11 downregulated (TMEM132B, MT3, H2-Q10, EXOC3L1, MUSTN1, SIM1, MYBPH, CLU, CRYAB, PEG3, STOM) in 3-week experiment tissue samples as compared to 1-week. The STRING analysis and functional enrichment analysis by the Geo Omnibus database revealed that upregulated genes belong to the biological processes of immune response and defense, adhesion, and complement, among others (Figure 9b). However, confirming differentially expressed genes, FN1 (data not shown) and C2, with qPCR in the whole group of mice did not confirm marked differences in gene expression between the 1-week model and the 3-week model.

### 3.7. Local Expression of the Components of the Complement Cascade in the Abdominal Wall of CHX-Induced Model Mice

To further investigate the expression of the complement cascade pathway in the abdominal wall of CHX-treated mice, we measured the gene expression of the complement molecules (C2, C3, C5, C6, CD55) in the CHX group compared to controls (Figure 10). C3, C2, and C6 were significantly expressed in the abdominal wall of the CHX-induced model as compared to control. We observed a slight trend (not significant) of increasing the gene expression of C2 and C6 genes from the 1-week to the 3-week time point (despite a significant upregulation of C2, according to RNA sequencing compared to 3-weeks vs. 1-week), and, additionally, an opposite trend in the expression of the complement inhibitory molecule CD55.

## 4. Discussion

We demonstrate that 1 week of CHX treatment (i.e., four injections in 1 week) causes degradation and the loss of mesothelial cells, local inflammation, damage to the underlying muscle layer, and the deposition of extracellular matrix, including collagen fibers (i.e., peritoneal fibrosis), which was assessed histologically. Using a predefined systematic approach to tissue sampling, we showed that the most intense structural changes in the parietal peritoneum are induced around the ventral parts of the abdominal cavity (linea alba, regio umbilicalis), very likely due to irritation of the injected CHX solution (gravity). The hyperspectral imaging of the entire area of the parietal peritoneum on the abdominal cavity showed the same location, intensity, and pattern of structural changes in the parietal peritoneum, as observed in histology. Hyperspectral imaging is a novel method that can evaluate structural and chemical changes in tissues based on molecular and physiological parameters of the tissue (i.e., blood, edema, collagen deposition, immune cell infiltration, etc.) using the light characteristics of penetration, absorption, reflection, transmission, and scattering (i.e., scattering parameters). It enables the evaluation of surface alterations as well as the degree of inflammation or fibrosis in the tissue at a depth of 1 mm [19].

However, inflammation and extracellular matrix deposition (signs of fibrosis) were also observed in the visceral peritoneum in a similar location-dependent pattern. Abdominal organs located ventrally were affected, while abdominal organs located dorsally, such as the kidneys and spleen, were not affected at 1 week of CHX treatment (dislocated from the CHX solution). Three weeks of CHX treatment led to multiplying the injury process and the progressive irruption of inflammatory cells into the parenchyma of the ventral parts of the abdominal organs (liver and intestine).

Similar results were found in all mice, which show that the CHX-induced model is robust and reproducible after both 1 and 3 weeks of CHX treatment. However, it is important that the CHX solution is always administered intraperitoneally. It is known that 11–13% of intraperitoneal administrations are injected into the intestine, subcutaneous, or fat. Therefore, in our study, intraperitoneal injections were performed by skilled persons in pairs (i.e., one person holding the mouse dorsally and the other injecting the needle into the lifted cone of the abdominal cavity.

Thus, our results show that the intensity and the thickness of the fibrosis depend on the location of the parietal peritoneum. Since the lesions in the CHX-induced model develop in a location-dependent manner (ventral abdomen), the sampling protocol thus significantly affects the results. Therefore, these results are important for future studies evaluating the potential protective effects of novel antifibrotic agents, such as drugs targeting TGF-β (pirfenidone), CTGF (pamrevlumab), phosphatidylinositol 3-kinase (parsaclisib), receptor tyrosine kinases (nintedanib), Janus kinase (ruxolitinib), and peroxisome proliferator-activated receptors (elafibranor), among others [27].

In many studies, the effect of agents on CHX-induced peritoneal fibrosis has been evaluated by the histological measurement of the thickness of the fibrotic alteration in the parietal peritoneum. However, it is important to note that the histological measurement of thickness was proposed for the encapsulating peritoneal sclerosis model, in which the changes are very thick. However, measuring the thickness of the fibrosis in relatively thin tissues (such as mouse peritoneum) can result in false results due to numerous technical errors of sampling and tissue processing. Tissues for histology can be sectioned in different planes, and if it is not sectioned perpendicularly but in a different plane (tangentially), this can lead to false differences between samples.

The molecular markers of fibrosis, such as the gene expression of COL1A1, ACTA2, and FN, increased markedly with the CHX stimulus but did not change with the number of CHX applications. Furthermore, the RNA sequencing of the abdominal wall tissue at 1 week and 3 weeks of the experiment revealed no significant differences in gene expression between the two time points.

The application of CHX successfully stimulated granulopoiesis, as we observed an increase in PMNs in the circulation that migrated into the abdominal wall and the liver. Locally (abdominal wall, liver), the percentage of PMNs decreased slightly from week 1 to week 3, which is consistent with the general timeline of inflammation, in which the initial infiltration of neutrophils is later replaced by other immune cells, such as macrophages [28]. We also observed the migration of PMNs from the tissue into the peritoneal cavity (measured in peritoneal lavage), which was significantly lower at 3 weeks compared to 1 week. This may be due to the loss of epithelial tight junctions reported with repeated 3-day CHX injections, which modulate the migration of PMNs through the epithelial layer [29,30]. Additionally, the accumulation of collagen over time, and thus the increased density and stiffness of the ECM upon CHX stimulation, significantly affects the migration of PMNs and slows their accumulation in the peritoneal cavity [31]. In accordance with the histological findings, the reports on the CHX-induced peritoneal fibrosis models [32], the loss of the mesothelial layer in the abdominal wall of the model mice was observed in the CHX-induced model. Peritoneal mesothelial cells have been shown to express both epithelial and mesenchymal markers and are capable of undergoing a mesothelial to mesenchymal transition under a proinflammatory/profibrotic stimulus and contributing to fibrosis via myofibroblasts [1]. The loss of mesothelial cells with their expression of chemotactic molecules [33] may also have an impact on the reduced recruitment of PMNs into the tissue.

CHX treatment increased several inflammatory markers in the abdominal wall at the level of gene expression (SAA1, SAA2, SAA3, IL6), similar to what was shown by Yokoi et al. (IL-1β, TNF-α) [15]. Expression was confirmed at the protein level for SAA3, which aligns with other inflammatory markers (TGF-β, SMAD2/3, MCP-1) presented previously [12,34]. Interestingly, however, the systemic circulating levels of inflammatory molecules, such as SAA (Figure 7d), IL-6, TNFα, IL-1b, IL-10, IL -1α (not shown as below detection limit), and others, were not affected by CHX at any time point. This is consistent with no reports of systemic inflammation in other CHX-induced peritoneal fibrosis mouse model studies. From a broader perspective, peritoneal dialysis patients who developed peritoneal sclerosis, characterized by thickening of the peritoneum, calcification, the presence of inflammatory elements, and angiogenesis, lack a systemic inflammatory response, as in our case [1].

We demonstrate that the gene expression of the complement molecules (C2, C3, C5, C6) and the inhibitory molecules (CD55) is increased in the abdominal wall of the CHX-induced mouse model of peritoneal fibrosis. This suggests that the complement system is locally expressed and possibly activated by the CHX inflammatory stimulus. The complement system (C1q, C3, C4, C5, CD9, MAC, CD46, CD55, CD59) has been studied in the human peritoneal membrane and is thought to play a role in peritoneal dialysis-related complications [35,36]. Although these proteins are mainly produced and secreted by the liver, local complement production and activation are necessary and important for the initiation of the immune response in serum-restricted sites. Immune cells can form a fully functional complement pathway in their microenvironment [37]. Mesothelial cells in the peritoneal cavity have been shown to express complement proteins and complement regulators [38,39], and large molecular size complement components have also been found in peritoneal dialysis effluent [40,41]. We observe a loss of the abdominal wall mesothelial layer in the CHX-treated group; therefore, the source of complement expression is more likely to be immune cells or other cells in the tissue. The inflammatory processes in the CHX-induced model could be related to the complement components through multiple pathways. The balance between CD46 (which inhibits the formation of C5a) and the decay accelerating factor CD55 (which promotes the formation of C5a) is important. C5a is a potent chemoattractant involved in the recruitment of inflammatory cells, including neutrophils, and can drive fibrosis by enhancing TLR2-mediated responses via the C5aR-TLR2 crosstalk [42]. In the CHX-treated group, there was an increase in local expression of the C2 component, a source of the C2b molecule (prokinin). Prokinin is then cleaved by plasmin to kinin, a molecule associated with edema [43,44], and may explain the greater edema formation at 3 weeks compared to 1 week in our model.

To summarize, in the CHX-induced model both histological and cellular/molecular changes are already present at week 1. Precise intraperitoneal administration and sampling protocol enable homogenous and reproducible lesions that can enable the investigation of the onset of peritoneal fibrosis, a process whose understanding may lead to better treatment strategies.

## 5. Conclusions

Our study on the CHX-induced peritoneal fibrosis mouse model confirms the previous findings of progressive fibrosis over time with repeated CHX injections. Considering the overall picture, we show that fibrosis affects not only the abdominal wall but also other abdominal organs. CHX-induced inflammation was characterized by both local and systemic increases in PMNs and local increases in SAA and IL-6 gene expression. However, no systemic increase in inflammatory markers was observed. We demonstrate for the first time the importance of the complement system in this model and emphasize its possible mechanistic involvement in the recruitment of PMNs and fibrosis. From the perspective of refining further experiments and thus limiting unnecessary animal suffering, we can conclude that 1-week of CHX treatment is sufficient to observe changes in inflammation and fibrosis, thus allowing the investigation of the underlying mechanisms and therapeutic agents at the onset of fibrosis. However, to obtain a reproducible model with low variability, precise intraperitoneal application and the use of an appropriate sampling protocol are recommended, and the results should take into the account the location of lesions.

## Figures and Tables

**Figure 1 biomedicines-10-02726-f001:**
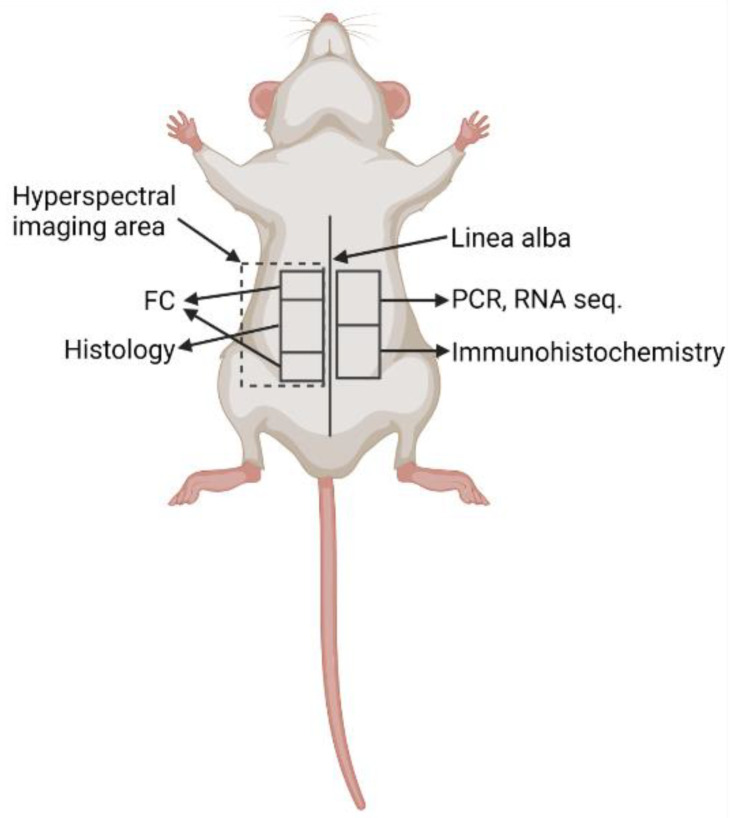
Locations of tissue collection for different methods were consistent in all animals. Legend: FC: flow cytometry, PCR: polymerase chain reaction, RNA seq: RNA sequencing. Created with BioRender.com.

**Figure 2 biomedicines-10-02726-f002:**
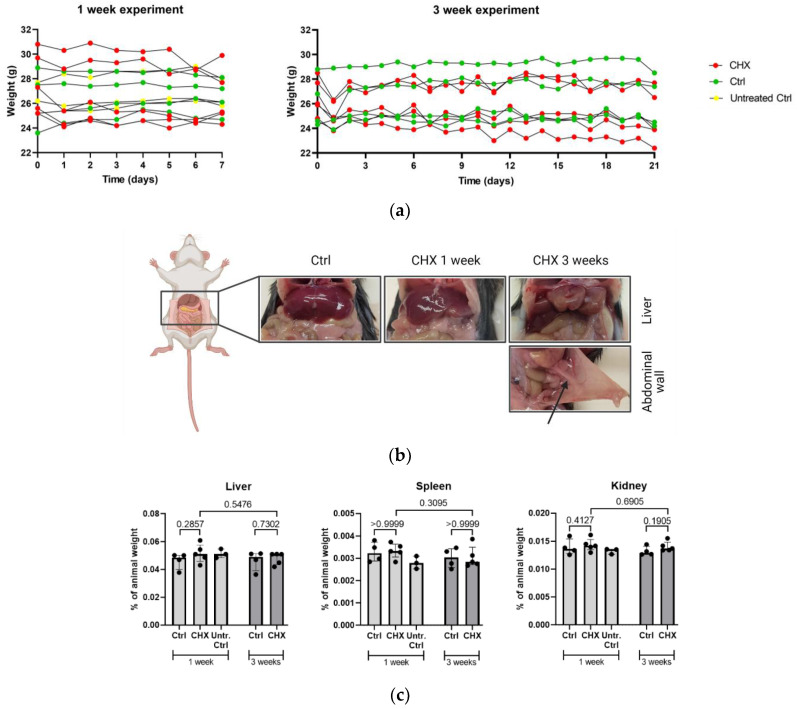
General observations on the CHX-induced mouse model. (**a**) The body weight of CHX-treated, treated control, and untreated control mice throughout the 1-week and 3-week experiment. Body weight of mice in 1-week experiment: no changes were seen in the group of untreated control mice (from 26.4 ± 1.3 g to 26.6 ± 1.2 g, *p* = 0.551 (*n* = 3)) and control mice (26.3 ± 2.4 g to 26.5 ± 1.5 g, *p* = 0.659 (*n* = 4)), while there was a decrease in the group of CHX-treated mice (27.7 ± 2.5 g to 26.5 ± 2.3 g, *p* = 0.024 (*n* = 5)). Similar results were observed when comparing the weight of mice in 3-week experiment; in control mice, weight did not change over time (from 26.1 ± 2.1 g to 26.2 ± 2.1 g, *p* = 0.921 (*n* = 4)), while there was a progressive weight loss over time in the group of CHX-treated mice (26.6 ± 1.5 g to 24.9 ± 2.159 g, *p* = 0.017 (*n* = 5)). (**b**) Macroscopic changes in the liver and the abdominal wall in the control, 1-week, and 3-week CHX-treated mice. Macroscopic changes in organs and tissues of the abdominal cavity were already seen at the 1-week time point and progressed severely to the 3-week time point. The arrow shows the attachment of mesentery to the abdominal wall present in 1/5 CHX-treated mice. Created with BioRender.com. (**c**) The relative weights of the liver, the spleen, and the kidney, normalized to the total animal weight, did not differ between the untreated controls, treated controls, and CHX-treated mice. Legend: CHX: chlorhexidine gluconate, ctrl: control, untr. ctrl: untreated control.

**Figure 3 biomedicines-10-02726-f003:**
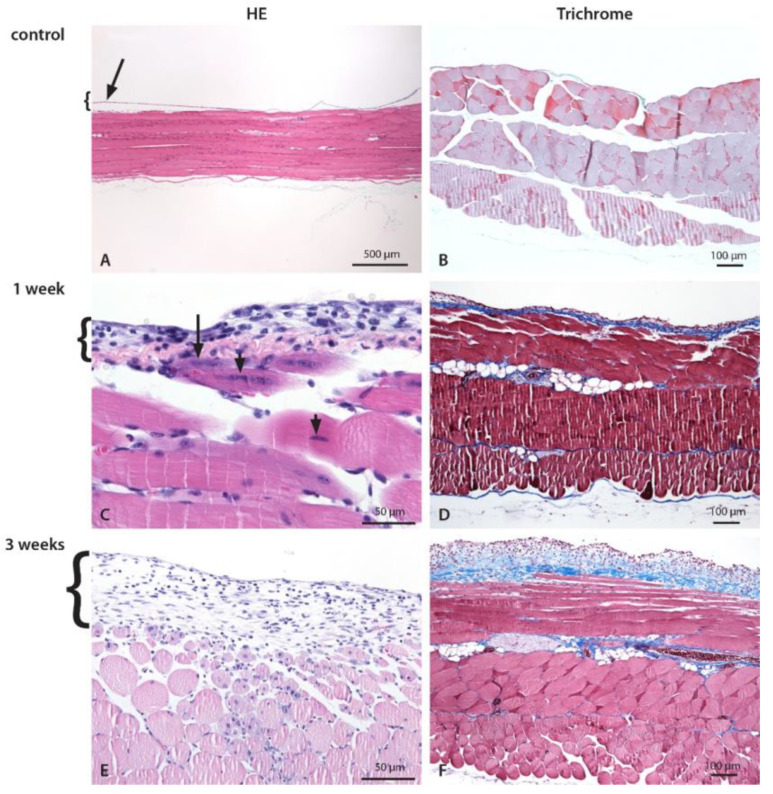
Histological changes of the abdominal wall and the parietal peritoneum after CHX treatment. Inflammation with mixed cell infiltrate composed of lymphocytes, macrophages, and neutrophils. Numerous necrotic muscle cells comprising up to 25% of the depth of the muscle wall infiltrated with inflammatory cells and an increased amount of intercellular fluid (edema) were present in both groups. Edema, fibroblasts, and consequent fibrosis were more pronounced in 3-week CHX-treated mice. The abdominal wall of control (**A**,**B**), 1-week (**C**,**D**), and 3-weeks (**E**,**F**) CHX-treated mice. (**A**,**B**). Normal abdominal wall with a longitudinal section of striated muscles covered by a thin layer of the parietal peritoneum, which is partially detached (technical artifact) (arrow). (**C**) On the peritoneal surface, there is a mild to moderate exudate composed of neutrophils, macrophages, lymphocytes, and scarce fibroblasts. The inner layer of the striated muscle cells, with necrotic (arrow) and damaged (arrowhead) striated muscle cells, and mild edema. (**D**) Early fibrosis is present. (**E**) On the peritoneal surface, there is a mild to moderate exudate composed of neutrophils, scarce mononuclear cells (macrophages, lymphocytes), and fibroblasts. Inflammatory cells invade the edematous abdominal wall and surround necrotic striated muscle cells encompassing up to 1/4 of the abdominal wall. Necrotic muscle cells are infiltrated with inflammatory cells and an increased amount of intercellular fluid (edema). (**F**) Edema and fibrosis on the peritoneal surface and in the abdominal wall.

**Figure 4 biomedicines-10-02726-f004:**
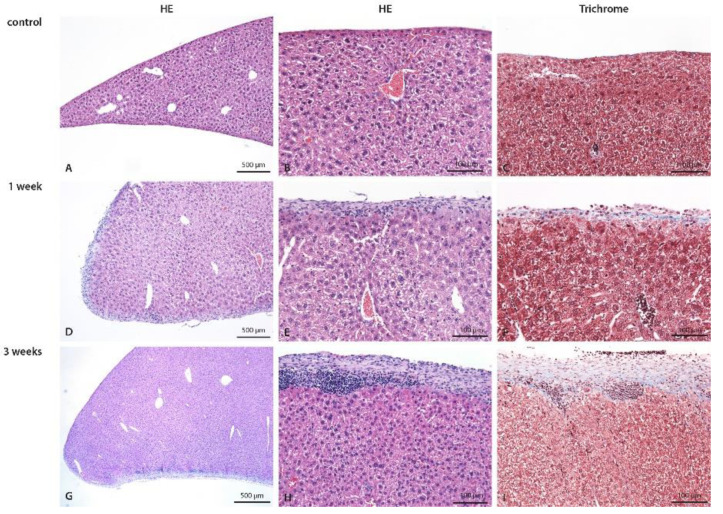
Liver of control (**A**–**C**), 1-week (**D**–**F**), and 3-weeks (**G**–**I**) CHX-treated mice with hematoxylin eosin and trichrome staining. In control mice, the normal liver is seen. In the 1-week CHX-treated mice edema of liver parenchyma, diffuse moderate exudate on the surface composed of mononuclear cells and fibrosis is present. In 3-weeks CHX-treated mice edema of liver parenchyma, diffuse moderate exudate on the surface composed of mononuclear cells and more intense fibrosis are present. Legend: HE: haematoxylin eosin, (**A**,**D**,**G**,**I**) 40×, (**B**,**C**,**E**–**H**) 100×.

**Figure 5 biomedicines-10-02726-f005:**
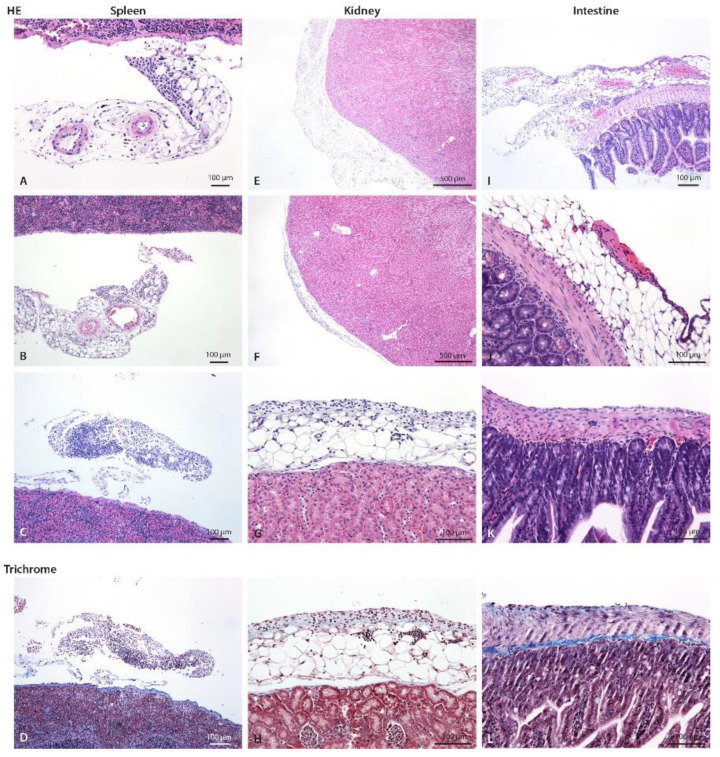
Spleen, kidney, and intestine of control and CHX-treated mice with hematoxylin eosin and trichrome staining. (**A**) Mild focal infiltrate composed of neutrophils and scarce lymphocytes and macrophages on perisplenic fat tissue/mesenterium (arrow) in control mice, (**B**) in 1-week and (**C**) 3-weeks CHX-treated mice. (**D**) Early fibrosis (blue). (**E**) A normal perirenal fat tissue at one pole (arrow) and (**F**) focal infiltrate on perirenal fat (arrow) at the other pole of the kidney in 3-weeks CHX-treated mice. Renal capsule and parenchyma are normal without inflammatory cells in the fat. (**G**) A closer view, infiltrate is composed of neutrophils, mononuclear cells, and fibroblasts. (**H**) Early fibrosis (blue). A similar infiltrate was present also in 1-week CHX-treated mice. (**I**) In the intestinal fat (subserosa) and mesenterium, there is mild to moderate infiltrate composed of neutrophils, macrophages, lymphocytes, and scarce fibroblasts, and focal fibrinous exudate on the surface of intestinal peritoneum consistent with fibrinous purulent peritonitis (arrow) in 3-weeks CHX-treated mice. (**J**) A closer view: scarce neutrophils and fibrinous exudate on intestinal serosa (arrow). (**K**) Mixed cell exudate on the intestinal wall. (**L**) Early fibrosis is present.

**Figure 6 biomedicines-10-02726-f006:**
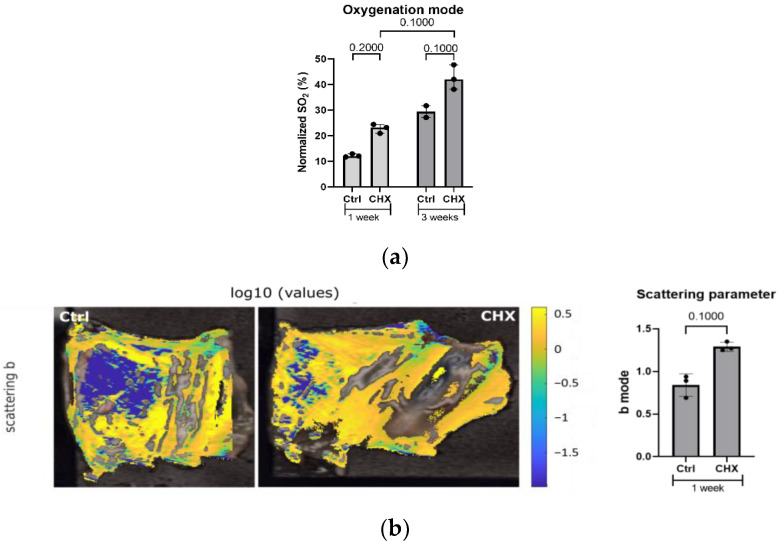
Hyperspectral imaging of the abdominal wall with parietal peritoneum. (**a**) Increased oxygenation (sample oxygenation mode value) in the vasculature was detected in the CHX-induced model compared to the control at both 1 and 3 weeks. Quantification was carried out on three mice/group. (**b**) Mode of scattering parameter was increased in the CHX-induced model at 1 week. In the uncolored sections, the measurement was not reliable enough for quantification, thus it was not calculated into mode. Quantification was performed on three mice/group. Legend: CHX: chlorhexidine, Ctrl: control.

**Figure 7 biomedicines-10-02726-f007:**
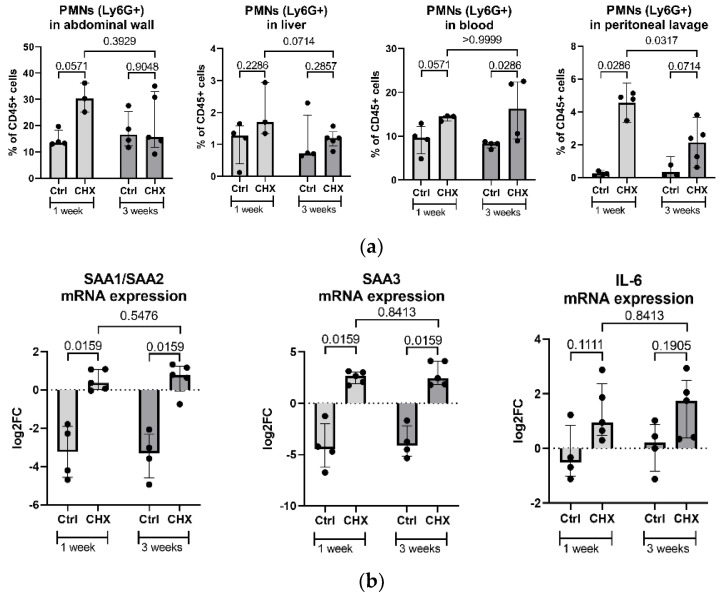
Inflammatory markers in the CHX-treated mice vs. controls. (**a**) Increased count of PMNs (shown as % of CD45+ cells) in the CHX-treated mice, both locally in the abdominal wall, liver, and abdominal cavity lavage and systemically in the bloodstream. (**b**) Gene expression of inflammatory markers SAA1/2, SAA3, and IL-6 in the abdominal wall is increased in the CHX-induced model but did not change regarding the time of the experiment. (**c**) Immunofluorescent labeling of the abdominal wall for SAA3 (green) and collagen IV (red) in 1-week CHX-treated mice shows positive cells for SAA3 in the tissue. (**d**) Serum levels of total SAA and SAA3. CHX treatment does not systemically affect the inflammatory markers SAA and SAA3, among others. Legend: CHX: chlorhexidine gluconate, Ctrl: control, PMNs: polymorphonuclear cells, SAA: serum amyloid A.

**Figure 8 biomedicines-10-02726-f008:**
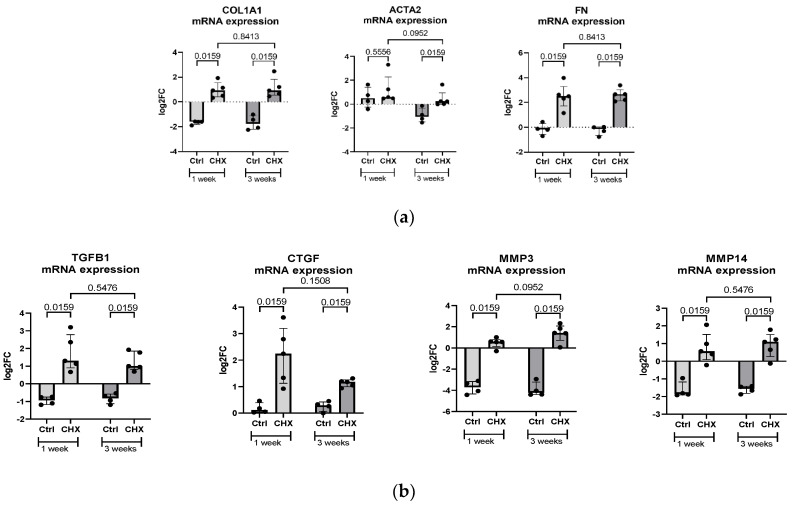
Fibrotic markers in the CHX-treated mice vs. controls. (**a**) Gene expression of profibrotic markers COL1A1, ACTA2, and FN in the abdominal wall. (**b**) Gene expression of the extracellular matrix regulators TGFB1, CTGF, MMP3, and MMP14 in the abdominal wall. Legend: CHX: chlorhexidine gluconate, Ctrl: control.

**Figure 9 biomedicines-10-02726-f009:**
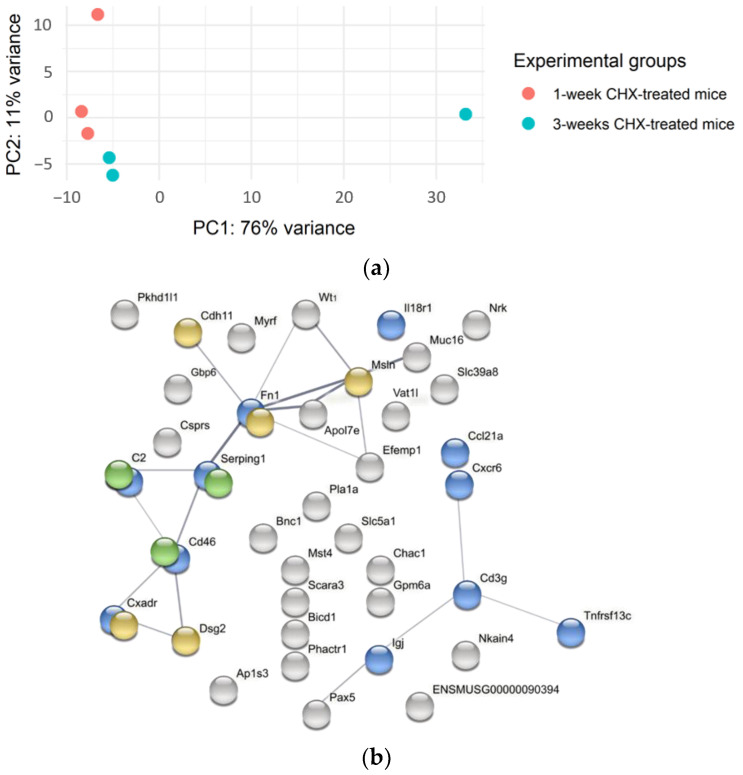
Gene expression heterogeneity as shown by PCA and clustering by STRING analysis as obtained from RNA sequencing of the abdominal wall of CHX-treated mice. (**a**) Principal component analysis shows the heterogeneity of gene expression within the 1-week (red) and the 3-week (blue) CHX-treated mice. (**b**) STRING analysis of a network of upregulated, differentially expressed genes in the abdominal wall between 3-week and 1-week CHX-treated mice. Network nodes represent proteins, edges represent protein–protein associations, both functional and physical, and line thickness indicates the strength of data support (thicker lines—more support). The minimum required interaction score is 0.4. The nodes are colored based on significant functional enrichments among those proteins as the most relevant biological processes (according to Gene Ontology): blue—immune response and defense; yellow—adhesion-related; and green—complement.

**Figure 10 biomedicines-10-02726-f010:**
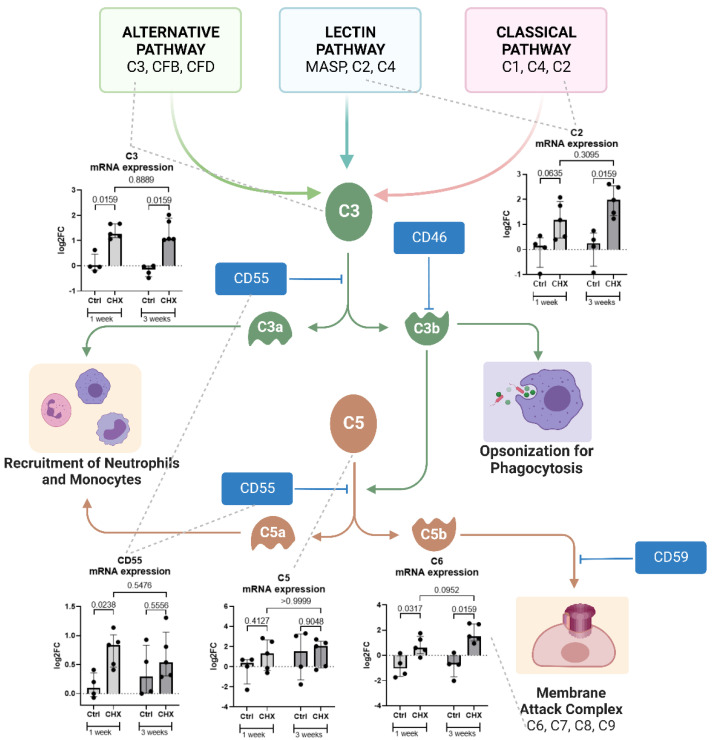
The complement pathway and increased gene expression of the complement cascade molecules C2, C6, C3, C5, and CD55 in the abdominal wall of the CHX-treated mice. Created with BioRender.com. Legend: CHX: chlorhexidine gluconate, Ctrl: control.

**Table 1 biomedicines-10-02726-t001:** Experimental protocol for the treatment of mice.

	1st Week	2nd Week	3rd Week
Group 1 Ctrl 1 week (*n* = 4)	200 μL PBS q.o.d. 1.6% EtOH in 100 μL PBS q.d.	/	
Group 2 CHX 1 week (*n* = 5)	200 μL 0.1% CHX in 15% EtOH in PBS q.o.d. 1.6% EtOH in 100 μL PBS q.d.		
Group 3 Ctrl 3 weeks (*n* = 4)	200 μL PBS q.o.d. 1.6% EtOH in 100 μL PBS q.d.		
Group 4 CHX 3 weeks (*n* = 5)	200 μL 0.1% CHX in 15% EtOH in PBS q.o.d. 1.6% EtOH in 100 μL PBS q.d.		
Group 5 Untreated Ctrl (*n* = 3)	No treatment	/	

Legend: CHX: chlorhexidine gluconate, ctrl: control, q.o.d.: every other day, q.d.: every day.

**Table 2 biomedicines-10-02726-t002:** Semiquantitative evaluation—histology sections of liver, spleen, kidney, and abdominal wall for immune cell presence and fibrosis.

Organ	Parameter	CHX		Control	
		1 Week	3 Weeks	1 Week	3 Weeks
liver	lymphocytes	++	+/++	-	-
	neutrophils	++	+/++	-	-
	fibrosis	+	++	-	-
spleen	lymphocytes	-/+	-/+	-	-/+
	neutrophils	+	++	-/+	-/+
	capsule	+	+	-/+	+
kidney	PMN/MNC	-/+	-/+	-	-
	fibrosis	-/+	-/+	-	-
Abd. wall peritoneum	PMNs	+/++	+/++	-	-
	MNCs	+	-/+	-	-
	edema	+/++	+++	-	-
	fibroblasts	+	++	-	-
Abd. wall injury	depth	up to 25%	up to 25%	-	-

No change (-), scarce (-/+), mild (+), moderate (++), severe (+++). CHX: chlorhexidine gluconate, PMNs: polymorphonuclear cells, MNCs: mononuclear cells.

## Data Availability

Data is contained within article and Appendix A.

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
