# Peer review of "Molecular and Cellular Markers in Chlorhexidine-Induced Peritoneal Fibrosis in Mice"

_biomedicines, 2022, doi:10.3390/biomedicines10112726_

Round 1

Reviewer 1 Report

The manuscript of Breznovec and colleagues aims to analyze which mediators are involved in the peritoneal fibroinflammation that characterizes the mouse model treated with chlorhexidine.

The work is interesting, well written and the data support the conclusions claimed by the authors. The methodologies are adequate for the intended purpose. Despite this, the paper should be strengthened with some experiments that could reinforce the message of this interesting work.

The main concerns are:

- The authors should better explain in the background how the model they use differs from other models of peritoneal fibrosis.

- The authors show that in various organs there is an accumulation of inflammatory infiltrate. The data are shown as a semi-quantitative score, but an immunohistochemical evaluation of the identified populations would be necessary to strengthen the data, using specific markers (e.g. F4 / 80 for macrophages, etc.).

- Similarly to what was said before, the expression of some inflammatory and fibrosis mediators (collagen 1A1, aSMA, TGFb1, etc) should be evaluated by immunohistochemistry in order to understand which cell types express them.

Reviewer 2 Report

The paper "Molecular and Cellular Markers in Chlorhexidine-Induced Peritoneal Fibrosis in Mice" is original with promising results. In order to make the results clearer it is required:

Line 44: Add referents

Line 48: many what? please, english language required

Line 48-56: add referents

Line 95: it is better to add the ethical permit and than experiment protocol

Row 98: Please add the weight of the animal

Line 110: Add the referents to the protocol used here

Line 129: then euthanasia ,?

Line 149: specific brand and type of microscope, please

Line 151: Specify reference references for the user protocol

Row 169: come up

Line 498: which ones? Indicate some of them

Line 532-538: explain better these results, also in relation to data already present in the bibliography

Round 2

Reviewer 1 Report

The paper is improved and could be worty of publication